# TextGraphBART: Unifying Graph and Text with Structure Token

## Abstract

We propose a novel encoding method called "Structure Token" to unify the processing and generation of both graphs and texts with a single transformer-based model. This method allows graphs with text labels to be generated by a series of tokens, enabling both graph and text data to be handled interchangeably. By utilizing structure tokens, our model learns a unified representation, enhancing the ability to process diverse data without requiring extra modules or models. Additionally, the model can be trained like most transformer models with simply cross-entropy loss. To demonstrate the effectiveness of our method, we introduce a pre-training scheme inspired by mBART but adapted to leverage structure tokens. Our model, named TextGraphBART, uses the same architecture as normal Transformer Encoder-Decoder models with small modifications on the input and output to accommodate structure tokens. The evaluations show that this approach achieves comparable results against baseline models of similar sizes on both text-to-graph and graph-to-text generation tasks, without needing specialized loss functions or sampling techniques. These findings suggest that our approach can effectively bridge the gap between textual and structural data representations, and the design of encoding method could offer a new direction for future improvement.

## 1 Introduction

Transformer layers have been proven to work well in several domains beyond text, like audio, image, and even multi-modal data. Some research has also shown that with careful design, transformer layers can extract features from graph data[30, 16]. Graph is a common data structure for representing concepts and relationships. In this work, we focus on a specific type, named text graph, where the concepts and relationships are expressed as texts, such as knowledge graphs and parsing trees. Learning vector representations and generating new text graphs are two essential aspects of text graphs in machine learning. Since texts can be viewed as a chain of words or characters, the text graph then becomes a nested graph. The complexity of handling such a nested graph leads to two major approaches for generating text graphs.

The first strand is the multi-stage approach which generates concepts, relationships, and texts in different steps [22, 12]. The process usually involves multiple models that generate different parts of the text graph. For example, Grapher [22] uses the T5 [24] pre-trained model to generate all the concepts and then uses a relation extraction model to predict the relationships between every two concepts. This approach requires the models to generate a complete graph despite the edge sparsity of the target text graph. Therefore, the model includes a special "no-relation" relationship and turns every graph into a complete graph thus requiring extra predictions. Since the relation extraction is

---

[1]code available at: https://github.com

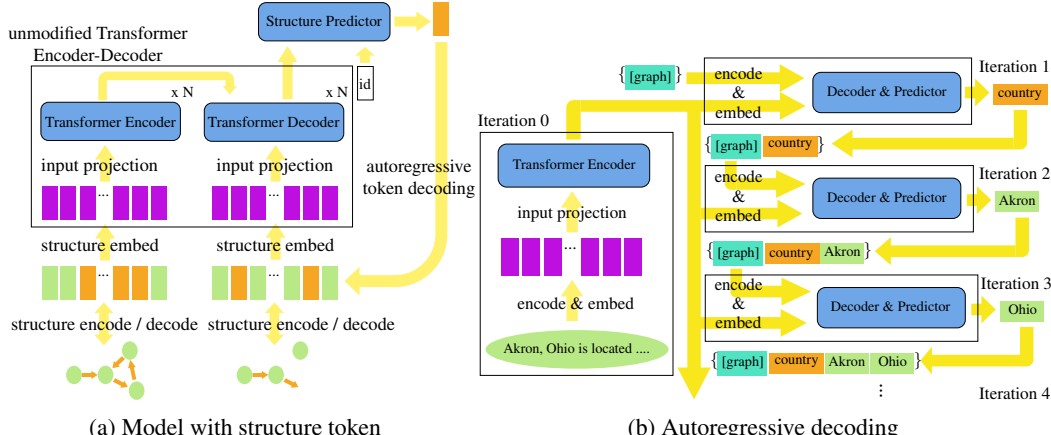

(a) Model with structure token

(b) Autoregressive decoding

Figure 1: Overview of the proposed structure token approach. (1a) The model takes the input text graph (left) and the partially generated sub-graph (right) and then generates a new structure token. Each structure token contains a (sub-)word token with the locational information of that word token in the text graph. (1b) An example of autoregressive decoding with structure token. The procedure is mostly the same as normal text decoding with Transformer model.

done on every two concepts, the model does not consider multi-hop relations. Moreover, the model cannot handle the case where two concepts have more than one relation. The second method is the graph linearization approach that fuses the hierarchy in text graph into chain of tokens [1, 10]. This approach treats the text graph as a special text sequence and enables the direct adoption of Language Model (LM) for text graph generation. The idea can also be applied to learning vector representations of text graphs. For example, BT5 [1] convert the text graph into sequence of (`subject, relation, object`) triples and train T5 to translate between sentence and sequence of triples. Since sequence generation with LM is done in an autoregressive manner, the generation is conditioned on the already generated triples. This behavior allows the model to handle multi-hop relations and the multi-relational case. Meanwhile, the model does not suffer from generating complete graphs because the model can learn to terminate when the generated triples match the target text graph. However, using sequence of triples also introduces extra complexity to the LM. Since the format requires matching the concepts in triples to reconstruct the text graph, there will be duplications of subject or object in the sequence. Thus, the model generates duplicated texts and cannot handle the case where two concepts are represented by the same text but refer to different things. Also, it relies on the model to implicitly learn the connection between two triples with duplication. Furthermore, LM is neither permutation invariance nor equivariance, which means the prediction alters if the generated triples are being shuffled.

The goal of this work is to design a new approach that preserves some of the advantages while avoiding the drawbacks of previous approaches. This sets a few desired properties of the new approach. First, the method should be suitable for both representation and generation. It should also consider the cases that cannot be handled by multi-stage and graph linearization approach. Second, the model should be permutation equivariance and perform generation in an autoregressive manner. Last, the method should avoid extra computation, such as the duplication of concepts. To achieve the desired characteristics, we propose the structure token approach, as illustrated in 1. Our method employs a concept we call "Structure Token" which losslessly encodes the text graph into a set of tokens. The token contains a word and a few identifiers for the precise location of that (sub-)word in the text graph. Our model incorporates an unmodified Transformer Encoder-Decoder model [26] with structure token embeddings and a structure predictor for predicting new structure tokens. The text graph is generated autoregressively like regular text generation and the generated structure tokens form a subgraph. Once the generation stops, we can decode the set of structure tokens into the target text graph. A notable difference between our approach and previous approaches is that our Transformer model operates on sets instead of sequences and we view text graphs as nested graphs. To our knowledge, our structure token approach is the first method that can autoregressively generate sub-graphs with multi-token labels without modifying transformers.

We validate our structure token approach on text-to-graph (T2G) generation and graph-to-text (G2T) generation tasks. By treating sentence as a special text graph without any edges, the generation tasks become a text-graph-to-text-graph translation problem. Therefore, we present TextGraphBART, a Transformer Encoder-Decoder model pre-trained on text-graph-to-text-graph translation with our structure token approach. The model is evaluated on two publicly available parallel datasets, EventNarrative [8] and WebNLG (2020) [5], and achieves comparable results using fewer parameters and pre-training data.

## 2 Related Work

Recent works for text graph generation primarily focus on reusing pre-trained LMs. BT5 [1] applies the graph linearization approach with T5 [24] pre-trained model. ReGen [10] further proposes a Reinforcement Learning (RL) objective to improve the performance. On the other hand, Grapher [22] uses T5 as an entity extraction model and jointly trains another relation extraction model. It provides two types of implementation: Grapher (Text) and Grapher (Query). The Grapher (Text), which is the state-of-the-art method on WebNLG (2020) [5] dataset, generates entities as a flat sequence, while Grapher (Query) feeds a set of query embeddings to T5 decoder and apply another model to fully decode the outputs to entities. Other works target on the problem of lacking paired datasets for T2G generation. CycleGT [12] apply the cycle-training framework on a G2T model and a multi-stage T2G model. INFINITY [29] apply the cycle-training framework on a single T5 model with graph linearization approach.

Most of the works for learning vector representations only focus on G2T generation. GAP [9] proposes a graph-aware attention that first uses the pooling operator to get the features of each text label and then applies a modified attention operator on those features. KGPT [6] provides two types of encoder: graph encoder and sequence encoder. The graph encoder is based on graph attention network [27] that also operates on the pooled text features. On the other hand, the sequence encoder infuses the structure information into the text tokens and feeds them into a transformer model. This approach resembles our structure token approach upon learning vector representations. However, they convert the graph into a dictionary-like format which suffers from a similar duplication problem as graph linearization approach.

Our method is essentially derived from the design of TokenGT [16], which also converts graphs into sets of tokens containing labels and identifiers. However, their idea does not directly fit in our scenario of text graph for two reasons. First, a single graph element (node or edge) in TokenGT needs to be representable by a single token. On the contrary, it would require multiple tokens for an element of text graph because the label is a multi-token text. Second, TokenGT only focuses on representing the graph, while we are interested in graph generation as well.

## 3 Method

### 3.1 Structure Token

The proposed structure token is a data representation that can losslessly encode all data in a text graph as a set of tokens. Given a text graph $\mathcal{G} = (\mathcal{N}, \mathcal{A})$ containing a node set $\mathcal{N}$ and an arc set $\mathcal{A}$. Each arc is a triple of a head node, an edge, and a tail node. Each graph element (node and edge) is a unique text label $\mathbf{S}$ identifiable with an integer ID. This allows different nodes or edges to have the same text label. The full formal definitions can be found in Appendix A.1. In order to convert text graph to structure tokens, we express the node set and arc set into one unified structure of graph elements. Each graph element will be represented by multiple structure tokens. A structure token consists of seven parts: 1. **Label:** The (sub-)word token of a graph element. 2. **Type:** A binary indicator specifying whether this graph element is a node or an edge. 3. **Token ID:** An unique ID for this token. 4. **Previous ID:** The token ID of previous token. 5. **Segment ID:** An unique ID for the graph element. 6. **Head ID:** The segment ID of the head node. 7. **Tail ID:** The segment ID of the tail node. If the token is part of a node, the head ID and tail ID will just be the segment ID of itself. With these information, we are able to differentiate between structure tokens that are from different parts of the graph. The text graph is converted into a set of structure tokens. Since the IDs can point to a graph element directly, there is no need for duplications like graph linearization approach. We provide the formal definition of structure token in the Appendix A.2. A real example of text graph

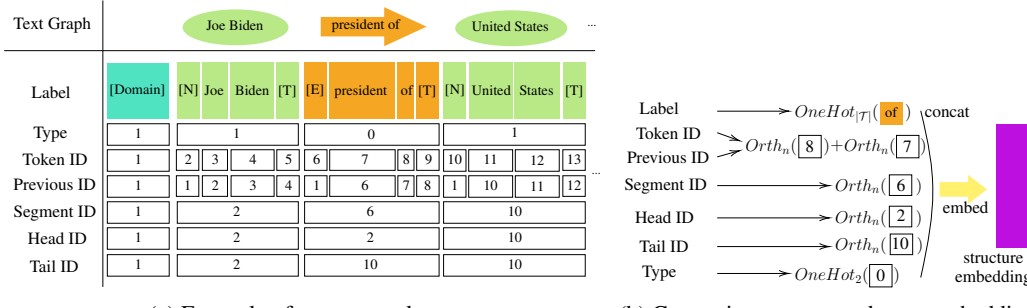

(a) Example of structure tokens.    (b) Converting structure token to embedding.

Figure 2: Structure token and embedding. (2a) Each column is a structure token and each token has a unique token ID. The IDs can be used to locate the (sub-)word in the text graph. (2b) Both $OneHot$ and $Orth$ convert the ID or word into a vector and those vectors would be concatenated together to form the embeddings of the structure token

and corresponding structure tokens can be found in Figure 2a. The idea of type, head ID, and tail ID are inherited from TokenGT [16], which uses identifiers to indicate the connections. We modify their definition and introduce extra identifiers for our text components. The token ID and previous ID are text-level identifiers. The text order is determined by the token ID and previous ID for reconstructing the text label. On the other hand, the segment ID, head ID, and tail ID are graph-level identifiers. For tokens of a specific graph element, the graph-level identifiers of each token will be the same.

Furthermore, We add an extra "domain token" to the structure tokens of a text graph to indicate the domain of the graph, like the special language token used in multi-lingual translation [15, 21]. With the domain token, we can specify what kind of data the text graph is holding. For example, since text is treated as a text graph without any edges, we use a "[text]" domain token to indicate that this text graph represents a text. Besides, we use the domain token as the first token of every text label, so the previous ID of the first token of all labels are pointing to the domain token.

## 3.2 Structure Embedding

The structure tokens are transformed into fixed-size high-dimensional vector representations. Each part of the structure token is converted into a vector and concatenated together. Then the vectorized result will be fed to a trainable projection layer for getting the token embedding, as illustrated in Figure 2b. Label and type are converted with one-hot encoding, denoted as $OneHot$. On the other hand, the IDs need to be handled differently. In order to preserve the graph structure in the tokens with the Transformer model, each ID needs to be converted into orthonormal vectors as proved by TokenGT [16]. We loose the requirement of orthonormality and use a set of orthonormal-like vectors. The dot product value of two different orthonormal-like vector is close to zero or less than some thresholds. These vectors of identifiers enable the attention operation in the Transformer model to be able to aggregate corresponding information through dot product. Each ID would be converted into an orthonormal-like vector through a transform function $Orth$. We use this transform function to convert the graph-level identifiers directly. On the other hand, we add the vectors of the text-level identifiers together, which allows the attention to aggregate information from neighbor tokens like the absolute position embedding [26]. Details and definitions can be found in Appendix A.3. Unlike position embedding which depends on the location of the tokens in a sequence, the text-level identifiers directly point to the neighbor token no matter their location in the sequence. Meanwhile, sequence orders are defined by IDs and the IDs can be randomly assigned. Therefore, applying any permutation of the input embeddings is equivalent to applying the same permutation on the output hidden states.

## 3.3 Generation

**Text Generation** After converting the structure tokens into embeddings, those embeddings are fed into the unmodified Transformer Encoder-Decoder model. Conceptually, our model generates a structure token at each step which contains seven objects. However, we do not really need to generate seven objects at every step. The token ID is unique for every token and we can randomly pick any ID sequence beforehand. Notably, the structure token representation is a set, while the autoregressive

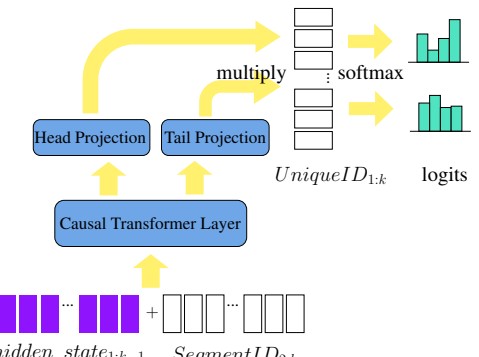

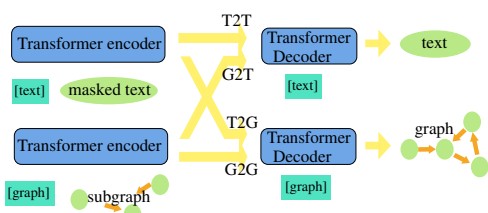

Figure 3: The Structure Predictor predicts graph-level identifier of the $k$-th token by taking the hidden state of previously generated tokens plus the segment ID of the next tokens. The output is then multiplied with all possible token IDs.

Figure 4: Illustration of our pre-training scheme. The input data is corrupted with masks or subgraph sampling on the encoder side, and the decoder would need to generate the uncorrupted data based on a domain token.

generation manner makes the generated tokens resemble a sequence. Although the design of structure tokens enables the possibility of non-monotonic order of text generation, we slightly restrict the generation order of the structure tokens from the same graph element to be ordered and contiguous. With this restriction, we do not need to predict the token ID and previous ID. We can use the same generation scheme of other text generation Transformer model that simply generates the next text token until we are done with this element. Meanwhile, since the graph-level identifiers are the same within a graph element, we only need to predict the graph-level identifiers for the first token of labels. The generation can be further simplified for a single sentence since the graph-level identifiers are merely the token ID of the first token. Thus, text generation with our structure token approach is almost the same as other Transformer-based text generation models.

**Text Graph Generation** For text graph generation, the same methodology applies. We use a structure predictor for predicting the identifiers. The graph-level identifiers are the same within a graph element. The prediction of graph-level identifiers can be done only one time per graph element. Moreover, the type and segment ID can also be omitted because we can tell the values once we get the head ID and tail ID. As a result, our structure predictor only needs to predict the head ID and tail ID. For predicting the IDs, we employ a single causal Transformer layer (a layer of the Transformer decoder), as illustrated in Figure 3. The causal Transformer layer takes the output of the Transformer model plus the transformed segment ID to produce a hidden vector. The hidden vector will be fed into two projection layers to get a prediction of the head ID and tail ID. To get the ID, we multiply the final hidden vectors with a list of our orthonormal-like vectors, and perform softmax on the multiplication result to get the predictions. With this setup, we can apply the same teacher forcing technique as other Transformer decoders, so the training process is also parallelized.

### 3.4 Pre-Training

We introduce a pre-training method for our model based on the mBART pre-trained model for multilingual text generation [21]. The pre-training method contains two types of training objectives: the self-supervised objective and translation-like objective, as illustrated in Figure 4. The translation-like objective forces the model to generate tokens depending on the domain token. The self-supervised objective allows us to utilize more datasets without paired data (e.g. plain text datasets or sample subgraphs from a large graph database). By using both kinds of objectives, the effective training data are doubled. Meanwhile, the model is encouraged to learn a more unified representation. With these objectives, we could utilize many different datasets to improve our model.

Table 1: Datasets statistics.

| | Size (# samples / uncompressed disk space / # tokens in texts / # tokens in graphs) |
|---|---|
| TEKGEN | 6.3 M / 1.5 GB / 218 M / 99 M |
| GenWiki | 750 K / 1.1 GB / 27 M / 11 M |
| EventNarrative | 180 K / 135 MB / 12 M / 4 M |
| WebNLG (2020) | 13 K / 16 MB / 399 K / 301 K |

Table 2: Performance of Graph-to-Text on EventNarrative.

| | # Params | BLEU / METEOR / BERTScore |
|---|---|---|
| T5-Base | 220 M | 12.80 / 22.77 / 89.59 |
| T5-Large | 770 M | 34.31 / 26.84 / 93.02 |
| BART-Base | 140 M | 31.38 / 26.68 / 93.12 |
| GAP | 153 M | **35.08 / 27.50** / 93.38 |
| TextGraphBART | **75 M** | 33.06 / 27.17 / **94.23** |

Table 3: Performance of Text-to-Graph on WebNLG (2020). The Grapher-small* is obtained by running the officially released source code of Grapher with T5-small weights. The # Params of CycleGT is not disclosed [5].

| | # Params | F1 (Strict / Exact / Partial) |
|---|---|---|
| CycleGT | N/A | 0.309 / 0.342 / 0.360 |
| BT5 | 770 M | 0.675 / 0.682 / **0.713** |
| Grapher (Query) | 770+ M | 0.289 / 0.395 / 0.325 |
| Grapher (Text) | 809 M | **0.681 / 0.683 / 0.713** |
| Grapher-small* (Text) | 95 M | 0.561 / 0.569 / 0.592 |
| TextGraphBART | **75 M** | 0.555 / 0.570 / 0.602 |

# 4 Experiments and Results

**Datasets** We use four parallel datasets containing both text and text graph for our experiments, as presented in Table 1. The model is pre-trained on TEKGEN [2] and GenWiki [14], and then we fine-tune the pre-trained model on EventNarrative [8] and WebNLG (2020) [5] for evaluating our model on G2T and T2G generation, respectively. The datasets are automatically generated by aligning texts with existing databases, except WebNLG (2020). TEKGEN is a large-scale dataset curated by aligning a subset of the Wikipedia text with Wikidata [28]. GenWiki is another large-scale dataset built on Wikipedia text. The text graphs are collected from DBpedia [3]. EventNarrative is an event-centric dataset that contains text graphs from the EventKG [11] and Wikidata. The text is also a subset of Wikipedia text. WebNLG (2020) is crowd-sourced dataset crafted by human annotators. The text graphs are collected from DBpedia, while the texts are manually written by annotators. It contains 16 categories in the training set and 19 categories (3 extra categories) in the test set. We use the official data split for all the datasets. As for the metrics, we used BLEU [23], METEOR [4], and BERTScore [31] to evaluate the G2T performance, and we use the official evaluation script of WebNLG (2020) to evaluate the T2G performance.

**Setups** Our model is trained in two phases: pre-training and fine-tuning. We initialize our model from scratch and perform the pre-training method. For pre-training, We use the RAdam optimizer [20] with a learning rate of 0.0001. The model is updated with an effective batch size of 256 and being trained for 5 epochs on a single A100 40GB GPU. All fine-tuning experiments are done on a single RTX 3090 24GB GPU. For fine-tuning on EventNarrative, we use the Lion optimizer [7] with a learning rate of 0.00001. The model is updated with an effective batch size of 128 and trained for 20 epochs. For fine-tuning on WebNLG (2020), we use the Adam optimizer [17] with a learning rate of 0.0001. The model is updated with an effective batch size of 128 and trained for 100 epochs.

We use an overall hidden size of 512 for our model. The unmodified Transformer encoder and decoder both have 6 layers. Each attention has 16 heads, and we use a hidden size of 32 for self attentions and 64 for cross attentions. The feed-forward layer in Transformer has an input and output hidden size of 512, and the intermediate hidden size is 2048. We use these numbers for the structure predictor as well. For the activation functions, we use the GELU activation function [13] for Transformers and hyperbolic tangent function for the projection layers of structure predictor. During pre-training, we apply the dropout [25] on the attention weights and the residual connections with a dropout rate of 0.1. The model weights are randomly initialized with a mean of 0 and a standard deviation of 0.02.

For data processing, we use the same subword tokenizer as T5 which uses the Unigram tokenization method [24, 18]. The tokenizer has a vocabulary of 32100 text tokens, which contain 32000 subword text tokens and 100 reserved special tokens. We use the reserved special tokens for our domain tokens. Each dataset is assigned with a corresponding domain token for the graph data, while all text data from different datasets share the same text domain token. The samples in each dataset are truncated with a maximum length of 128 or 256 text tokens depending on the training stage. A random unique ID sequence is determined for each sample at every epoch. During the pre-training, we randomly

Table 4: Performance of our model on each category of WebNLG (2020) test set comparing to Grapher-small. The * denotes the unseen categories.

| Category | # Samples (train / test) | TextGraphBART (Strict / Exact / Partial) | Grapher-small (Text) (Strict / Exact / Partial) |
|---|---|---|---|
| Total | 13211 / 2155 | 0.555 / **0.570** / **0.602** | **0.561** / 0.569 / 0.592 |
| Airport | 1085 / 111 | 0.798 / 0.799 / 0.801 | **0.831** / **0.831** / **0.833** |
| Artist | 1222 / 129 | 0.636 / 0.650 / 0.666 | **0.696** / **0.709** / **0.727** |
| Astronaut | 529 / 102 | 0.797 / 0.805 / 0.809 | **0.847** / **0.847** / **0.848** |
| Athlete | 903 / 68 | 0.632 / 0.635 / 0.641 | **0.732** / **0.732** / **0.734** |
| Building | 972 / 53 | 0.811 / 0.812 / 0.817 | **0.889** / **0.889** / **0.890** |
| CelestialBody | 634 / 63 | **0.713** / **0.716** / **0.716** | 0.664 / 0.664 / 0.669 |
| City | 1110 / 104 | **0.565** / **0.580** / **0.588** | 0.387 / 0.390 / 0.395 |
| ComicsCharacter | 285 / 35 | 0.781 / 0.781 / 0.787 | **0.917** / **0.917** / **0.919** |
| Company | 351 / 93 | 0.878 / 0.878 / 0.880 | **0.919** / **0.919** / **0.919** |
| Film* | **0** / 333 | **0.338** / **0.391** / **0.439** | 0.260 / 0.284 / 0.323 |
| Food | 1406 / 51 | 0.756 / 0.761 / 0.761 | **0.908** / **0.908** / **0.908** |
| MeanOfTransportation | 1132 / 65 | **0.623** / **0.628** / **0.647** | 0.585 / 0.587 / 0.590 |
| Monument | 263 / 53 | 0.848 / 0.848 / 0.848 | **0.915** / **0.915** / **0.916** |
| MusicalWork* | **0** / 355 | **0.244** / **0.257** / **0.354** | 0.163 / 0.174 / 0.247 |
| Politician | 1194 / 34 | 0.805 / 0.805 / 0.807 | **0.810** / **0.810** / **0.811** |
| Scientist* | **0** / 302 | **0.499** / **0.510** / **0.538** | 0.483 / 0.490 / 0.516 |
| SportsTeam | 782 / 51 | 0.689 / 0.691 / 0.696 | **0.856** / **0.856** / **0.862** |
| University | 406 / 107 | **0.636** / **0.640** / **0.674** | 0.627 / 0.629 / 0.659 |
| WrittenWork | 937 / 46 | **0.400** / **0.405** / **0.526** | 0.297 / 0.308 / 0.384 |

assign a unique ID sequence with a maximum value of 512. For the encoder input, we randomly drop 15% of graph elements or tokens depending on the domain.

**G2T Results** We compared our model with T5 [24], BART [19], and GAP [9]. Both T5 and BART are Transformer Encoder-Decoder models pre-trained on text data and fine-tuned with graph linearization [8], while GAP modifies the encoder of Transformer Encoder-Decoder model with graph-aware modules for extracting graph features [9]. It is noteworthy that all these models use a similar Transformer decoder. The main difference among TextGraphBART and these models is the way we represent and handle the text graph input.

The result is shown in Table 2. In comparison to T5 and BART, our structure token method achieves better score with fewer parameters than graph linearization approach. Meanwhile, our model is comparable with GAP without modifying the Transformer model. As a conclusion, our structure token representations enabled the Transformer model to capture better features from the text graph than the graph linearization approach.

**T2G Results** We compare our model with CycleGT [12, 5], BT5 [1], and Grapher [22]. BT5 is T5 pre-trained and fine-tuned with graph linearization. On the other hand, both CycleGT and Grapher adopt the multi-stage approach. The CycleGT is a well-known multi-stage approach for text-to-graph generation using cycle training [12], while Grapher performs supervised learning with a special loss function [22]. Meanwhile, we use the officially released source code of Grapher[1] to train a Grapher-small (Text) which has a similar model size (95M) with our model (75M). Both Grapher-small and our TextGraphBART are trained for 100 epochs with the same learning rate and effective batch size.

The result is shown in Table 3. In comparison to CycleGT and Grapher (Query), our simple generation method with structure tokens outperforms models with special training methods. Although our model does not directly match the performance of the large models like BT5 or Grapher (Text), our model is comparable with Grapher-small that has similar model size. Furthermore, we analyze the result by measuring the performance on each category of the WebNLG (2020) test set comparing to Grapher-small. The result is shown in Table 4. Even though Grapher-small is based on the T5-small pre-trained model, which is trained on an extremely large dataset of 750 GB: the Colossal Clean Crawled Corpus

---

[1] https://github.com/IBM/Grapher

Table 5: Ablation results of our structure embedding on WebNLG (2020) test set.

| | F1 (Strict / Exact / Partial) |
|---|---|
| TextGraphBART | 0.555 / 0.570 / 0.602 |
| w/o segment ID | 0.547 / 0.562 / 0.595 |
| w/o type | 0.544 / 0.561 / 0.594 |
| w/o head ID & tail ID | 0.489 / 0.507 / 0.539 |
| w/o token ID & previous ID | 0.365 / 0.378 / 0.404 |

(C4) [24], we can see that our model performs slightly better than Grapher-small on unseen categories (0 samples in training set). In conclusion, our structure token approach can achieve comparable performance on text-to-graph generation under similar model size without using special training methods or loss functions.

**Ablation Study**   To investigate the performance contribution of the components of structure tokens, we conducted the ablation study on our structure embedding by fine-tuning our model with the removal of some parts of the embeddings. The model is trained on the WebNLG (2020) with the same setup. The results are shown in Table 5. In all ablations, the model performance was attenuated as expected. First, the ablation of the token ID and previous ID removes the text order information in the text labels hence the degeneration of performance. Similarly, the head ID and tail ID provide the connectivities of the graph. Removal of this embedding decreases the performance, indicating the importance of the connectivities. On the other hand, the ablation of type and segment ID are not as detrimental as others because the type and segment ID may be inferred from other IDs. Thus, our model is still able to perform albeit less performant. In conclusion, the ablation study showed that all of our structure embedding is important for good model performance.

# 5   Discussion

The primary objective of this work is to demonstrate the effectiveness of the proposed structure token approach. Due to resource constraints, there are numerous aspects remain unexplored.

**Scaling Up**   We believe that TextGraphBART has the potential to achieve better results when scaled up. The backbone of our model is simply the transformer model, like most of the other baselines. Meanwhile, we observed a clear improvement on other baselines when scaling up (e.g. comparing T5-Base and T5-Large for G2T in 2 and Grapher-small and Grapher for T2G in 3). Therefore, we anticipate that transformers on T2G and G2T follow the scaling laws.

**Model Architecture**   While we use The Transformer Encoder-Decoder model in our experiments, there are no strict restrictions on the model architecture as long as there are dot product attention operations. The same approach can be applied to encoder or decoder-only model.

**Data and Objective for Pre-Training**   In the experiments, we only use parallel datasets for pre-training. Since our pre-training scheme contains a T2T path, it's possible to pre-train the model merely with plain-text datasets. On the other hand, we can also incorporate programming language datasets and use syntax parsers to generate the Abstract Syntax Tree (AST) as the text graph for pre-training. However, it is unclear how much the self-supervised objective and translation-like objective affect the downstream performances.

# 6   Conclusions

We present a novel approach to the problem of text graph generation leveraging the strength of Transformer models. Our exploration has led to an effective method for structured data representation and generation via structure tokens. In the structure token, we use several identifiers to indicate the connectivities of the graphs and the order of the texts. Then an embedding method for structure token is proposed, allowing the Transformer model to utilize the structural information. We show that the structure token approach can be used to represent and generate both texts and text graphs.

The experiment results demonstrated the effectiveness of our method with less data and parameters. Meanwhile, the ablation study further confirmed the importance of various elements of the structure tokens.

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

# A  Formal Definitions

## A.1  Preliminaries

Let $\mathcal{T}$ be the set of all possible text tokens. A text label is defined as a set of pairs containing the text token and positions. We generalize this definition by replacing the position with a contiguous sequence of unique ID. Given an infinite unique ID sequence: $\mathcal{I} = (\mathrm{id}_i)_{i=0}^{\infty}$ where $\mathrm{id}_i \in \mathbb{Z}^+ \wedge \mathrm{id}_i \neq \mathrm{id}_j$ if $i \neq j$. Each $\mathrm{id}_i$ is a positive integer. We also define $\mathrm{id}(i) = \mathrm{id}_i$ for simplicity. By picking a corresponding ID sequence, we can use any positive integer sequence as the positions. Then a text label $\mathbf{S}$ of length $l$ is a set of token-ID pairs defined as: $\mathbf{S} = \{(t_i, \mathrm{id}_{j+i}) \mid 1 \leq i \leq l, t_i \in \mathcal{T}\} \subseteq \mathcal{T} \times \mathbb{Z}^+$. We can conditionally specify the start point $j \in \mathbb{N}$ with $\mathbf{S}^j$. Let $\mathcal{S}$ be the set of all possible text labels and $\mathcal{S}^j$ for a specific start point. With the ID sequence $\mathcal{I}$, the positions $(p)_{p=1}^{l}$ can be replaced with $(\mathrm{id}_{j+k})_{k=1}^{l}$. Then we can union text labels without missing information by picking non-overlapped ID sequences, which is a desired property for the attention operation. A text graph $\mathcal{G} = (\mathcal{N}, \mathcal{A})$ is composed of a node set $\mathcal{N}$ with $q$ nodes and an arc set $\mathcal{A}$ with $r$ arcs. The node set $\mathcal{N}$ is a set of node labels paired with unique IDs of the nodes, defined as: $\mathcal{N} = \{(N_i, n_i) \mid 1 \leq i \leq q, n_i \in \mathbb{Z}^+, N_i \in \mathcal{S}\} \subseteq \mathcal{S} \times \mathbb{Z}^+$ where $N_i$ is the node label and $n_i$ is the corresponding node ID. Similarly, an edge set $\mathcal{E}$ is a set of edge labels paired with unique IDs, defined as: $\mathcal{E} = \{(E_i, e_i) \mid 1 \leq i \leq r, e_i \in \mathbb{Z}^+, E_i \in \mathcal{S}\} \subseteq \mathcal{S} \times \mathbb{Z}^+$ where $E_i$ is the edge label and $e_i$ is the edge ID. Notably, the ID used in $\mathcal{N}$ and $\mathcal{E}$ are disjoint. Then the arc set is defined as: $\mathcal{A} = \{(\mathbf{N}_i^h, \mathbf{E}_i, \mathbf{N}_i^t) \mid 1 \leq i \leq r, \mathbf{N}_i^h, \mathbf{N}_i^t \in \mathcal{N}, \mathbf{E}_i \in \mathcal{E}\} \subseteq \mathcal{N} \times \mathcal{E} \times \mathcal{N}$ where $\mathbf{E}_i$ is the edge and $\mathbf{N}_i^h, \mathbf{N}_i^t$ is the head node and tail node, respectively.

 **A.2 Structure Tokens**

With the setup in A.1, the formal definition of the structure token representation is defined as a set of septuples (tuples with 7 elements). Given a text graph $\mathcal{G} = (\mathcal{N}, \mathcal{A})$ and its components: node $\mathbf{N}_i = (N_i, n_i) \in \mathcal{N}$, $1 \leq i \leq |\mathcal{N}|$, arc $\mathbf{A}_j = ((N_j^h, n_j^h), (E_j, e_j), (N_j^t, n_j^t)) \in \mathcal{A}$, $1 \leq j \leq |\mathcal{A}|$, edge $\mathbf{E}_j \in \mathcal{E} = \{(E_j, e_j) \mid 1 \leq j \leq |\mathcal{A}|\}$, and the length of the text labels $l_i^N = |N_i|$, $l_j^E = |E_j|$, we define two sequences:

$$\mathcal{L}^N = (\mathbf{L}_k^N)_{k=1}^{|\mathcal{N}|} \text{ where } \mathbf{L}_k^N = \begin{cases} 0, & \text{if } k = 1 \\ l_{k-1}^N + \mathbf{L}_{k-1}^N, & \text{if } k \neq 1 \end{cases}$$

$$\mathcal{L}^E = (\mathbf{L}_k^E)_{k=1}^{|\mathcal{A}|} \text{ where } \mathbf{L}_k^E = \begin{cases} \sum_{k=1}^{|\mathcal{N}|} l_k^N, & \text{if } k = 1 \\ l_{k-1}^E + \mathbf{L}_{k-1}^E, & \text{if } k \neq 1 \end{cases}$$

We assign $n_i = \mathrm{id}(\mathbf{L}_i^N + 1)$, $e_j = \mathrm{id}(\mathbf{L}_j^E + 1)$ and specify the start point of each text label such that $N_i \in \mathcal{S}^{\mathbf{L}_i^N}$, $E_j \in \mathcal{S}^{\mathbf{L}_j^E}$. By doing so, we can get the node (or edge) ID from the token IDs of its text label. For each node $\mathbf{N}_i$ and edge $\mathbf{E}_j$, we define the corresponding structure token representation $X_i^N$ and $X_j^E$ as:

$$\begin{aligned} X_i^N = \{(t_k, 1, uid_k, uid_{k-1}, n_i, n_i, n_i) \mid \\ 1 \leq k \leq l_i^N, (t_k, uid_k) \in N_i, uid_0 = \mathrm{id}_0\} \\ X_j^E = \{(t_k, 0, uid_k, uid_{k-1}, e_j, n_j^h, n_j^t) \mid \\ 1 \leq k \leq l_j^E, (t_k, uid_k) \in E_j, uid_0 = \mathrm{id}_0\} \end{aligned} \tag{1}$$

Then the corresponding structure token representation $\mathcal{G}'$ of the text graph $\mathcal{G}$ is defined as:

$$\begin{aligned} \mathcal{G}' = \bigcup_{k=1}^{|\mathcal{N}|} X_k^N \cup \bigcup_{k=1}^{|\mathcal{A}|} X_k^E \cup X_D \\ \subseteq \mathcal{T} \times \{0,1\} \times \mathbb{Z}^+ \times \mathbb{Z}^+ \times \mathbb{Z}^+ \times \mathbb{Z}^+ \times \mathbb{Z}^+ \end{aligned} \tag{2}$$

where $t_D \in \mathcal{T}$, $\mathrm{id}_0$ is the ID of domain token and $X_D = \{(t_D, 1, \mathrm{id}_0, \mathrm{id}_0, \mathrm{id}_0, \mathrm{id}_0, \mathrm{id}_0)\}$ is the domain token. Each septuple $X \in \mathcal{G}'$ is a structure token containing the label, type, token ID, previous ID, segment ID, head ID, and tail ID. With this definition, we can represent every possible token ID assignment by specifying the unique ID sequence $\mathcal{I}$. On the other hand, since the graph element can be randomly permuted, every possible ordering is also representable with our set $\mathcal{G}'$ by picking the corresponding unique ID sequence.

**A.3 Structure Embedding**

To convert the structure tokens into embeddings, we use 3 kinds of transform functions. For label and type, we use the one-hot encoding, denoted as $OneHot_n \colon \mathbb{A} \to \mathbb{E}_n$ where $\mathbb{A}$ is a set with $n$ elements and $\mathbb{E}_n$ is the standard basis of $\mathbb{R}^n$. On the other hand, each ID would first be converted into a $d$-dimensional orthonormal-like vector with a function $Orth_d$. To get the orthonormal-like vectors, we modify and normalize the sinusoidal position encoding of Transformer [26] with different frequencies. The $d$-dimensional sinusoidal position encoding $\mathrm{PE}_d$ at position $i$ is defined as:

$$\mathrm{PE}_d(i) = \Big\|_{k=1}^{d/2} pe(i, k) \quad \in \mathbb{R}^d \tag{3}$$

$$pe(i, k) = \sin(\frac{i}{10000^{k/d}}) \| \cos(\frac{i}{10000^{k/d}}) \quad \in \mathbb{R}^2$$

where $\|$ denotes the vector concatenation. We generalize the definition of $\mathrm{PE}_d$ with a frequency function $f$:

$$\mathrm{PE}_d^*\{f\}(i) = \Big\|_{k=1}^{d/2} pe^*\{f\}(i, k) \quad \in \mathbb{R}^d \tag{4}$$

$$pe^*\{f\}(i, k) = \sin(i * f(k)) \| \cos(i * f(k)) \quad \in \mathbb{R}^2$$

Then by taking $f'(k) = 10000^{-k/d}$, the original $\mathrm{PE}_d$ can be defined with $\mathrm{PE}_d = \mathrm{PE}_d^*\{f'\}$. We use this generalized position encoding to define the function $Orth_d \colon \mathbb{Z}^+ \to \mathbb{R}^d$ of the IDs for our

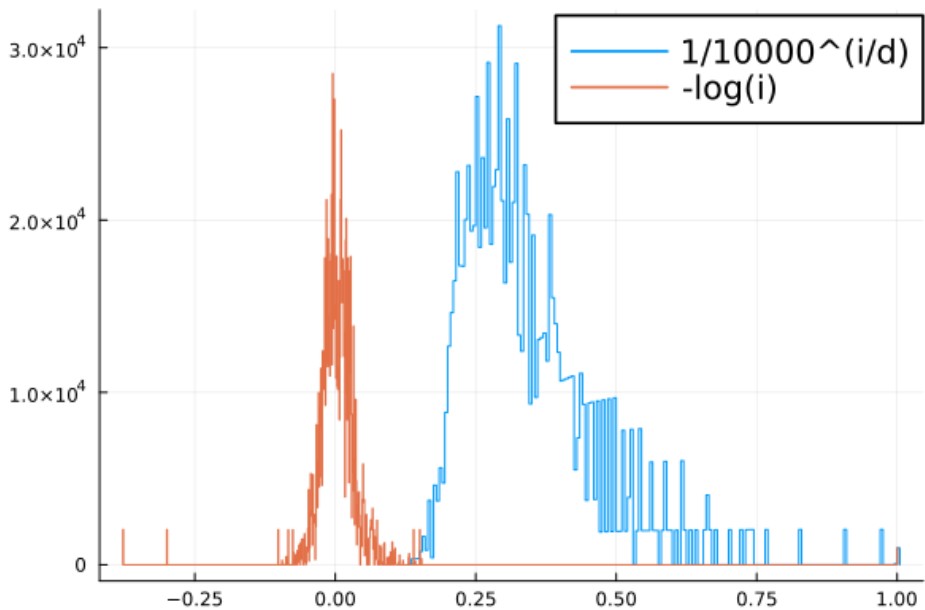

Figure 5: Histogram of cos-similarities of 1024 vectors with different frequency functions. The orange line is our $Orth$, while the blue line is the normalized sinusoidal position embedding. A dimensionality of 512 is used in this figure.

orthonormal-like vectors as:

$$Orth_d = norm_2 \circ \text{PE}_d^*\{-\log(k)\} \tag{5}$$

where $norm_2$ is the L2 normalization and $\circ$ denote the function composition. We find that by picking the frequency function $-\log(k)$, the generated vectors satisfied the desired properties. In Figure 5, we generate 1024 vectors by applying $Orth$ on $1 \le i \le 1024$ and compute the cosine similarity of every possible pair. We can see that the similarity values are mostly close to zero.

Once we can convert the IDs into orthonormal-like vectors, we use $Orth_d$ directly as the transform functions for the graph-level identifiers. For text-level identifiers, we add the vector of token ID and previous ID together. Given two non-domain token $a$ and $b$ with token ID $t_a$, $t_b$ and previous ID $p_a$, $p_b$, we have $Orth(t_a) \cdot Orth(t_b) \approx 0$ since token ID is unique and $Orth$ is orthonormal-like. Meanwhile, we have $p_a \neq p_b$ for most tokens. The dot product value of $a$ and $b$ become:

$$
\begin{aligned}
&(Orth(t_a) + Orth(p_a)) \cdot (Orth(t_b) + Orth(p_b)) \\
&= Orth(t_a) \cdot Orth(t_b) + Orth(t_a) \cdot Orth(p_b) + Orth(p_a) \cdot Orth(t_b) + Orth(p_a) \cdot Orth(p_b) \\
&= Orth(t_a) \cdot Orth(p_b) + Orth(p_a) \cdot Orth(t_b) + Orth(p_a) \cdot Orth(p_b) + \varepsilon \\
&= \begin{cases}
Orth(t_a) \cdot Orth(p_b) + \varepsilon = 1 + \varepsilon, & \text{if } a \text{ is the previous token of } b \\
Orth(p_a) \cdot Orth(t_b) + \varepsilon = 1 + \varepsilon, & \text{if } b \text{ is the previous token of } a \\
Orth(\text{id}_0) \cdot Orth(\text{id}_0) + \varepsilon = 1 + \varepsilon, & \text{if both } a \text{ and } b \text{ are the first token of text label} \\
\varepsilon, & \text{otherwise}
\end{cases}
\end{aligned}
\tag{6}
$$

where $\varepsilon$ is a small value around $0$. The value will only be meaningful in the dot product if one token is the previous token of the other. The case of first tokens is set to allow exchanging information with the domain token. This can be suppressed by making $Orth(\text{id}_0) = \mathbf{0}$.

With these designs, we define our structure token vectorize function $Vec$ as:

$$
\begin{aligned}
Vec(X) = \\
OneHot_{|\mathcal{T}|}(\pi_1(X)) \parallel OneHot_2(\pi_2(X)) \parallel (Orth_n(\pi_3(X)) + Orth_n\pi_4(X))) \\
\parallel Orth_n(\pi_5(X)) \parallel Orth_n(\pi_6(X)) \parallel Orth_n(\pi_7(X)) \in \mathbb{R}^{4n+|\mathcal{T}|+2}
\end{aligned}
\tag{7}
$$

where $\pi_i(X)$ denote the $i$-th element of the septuple $X$. The vectorized result will be fed into a trainable projection layer $Emb\colon \mathbb{R}^{4n+|\mathcal{T}|+2} \to \mathbb{R}^d$ to get the structure embedding with $d$ dimensions.

