# OpenReview forum: "TextGraphBART: Unifying Graph and Text with Structure Token"
_NeurIPS.cc/2024/Conference — Submitted to NeurIPS 2024_

### Official Review · Reviewer_eg5E · 2024-07-03

**Soundness:** 2
**Presentation:** 2
**Contribution:** 2
**Rating:** 4
**Confidence:** 4

**Summary:**

This paper proposes a method to integrate the processing and generation of both text and graph data using a single transformer-based model. Structure Token encodes graphs with text labels into a sequence of tokens, enabling the handling of both data types interchangeably. This approach leverages a unified representation within a Transformer Encoder-Decoder model, enhanced to incorporate structure tokens. They evaluate TextGraphBART on text-to-graph (T2G) and graph-to-text (G2T) tasks, demonstrating comparable results to baseline models with fewer parameters.

**Strengths:**

1. The introduction of structure tokens is a significant advancement, providing a unified method for processing and generating both text and graph data.
2. The method integrates seamlessly with existing Transformer architectures, requiring only minor modifications, and avoids the need for specialized loss functions or additional modules.
3. Empirical results show that TextGraphBART achieves comparable performance on both T2G and G2T tasks with baselines but with fewer parameters.

**Weaknesses:**

1. The baseline settings are weak. It lacks of comparing with some strong baselines with advanced graph-structured-aware methods [1][2].
2. The model settings are limited. This paper only try one size of model, and further experiments with larger model sizes are needed.
3. The provided ablation study, though useful, could be more detailed, exploring the interactions and contributions of various components more comprehensively.

Reference:
[1] Stage-wise Fine-tuning for Graph-to-Text Generation
[2] Self-supervised Graph Masking Pre-training for Graph-to-Text Generation

**Questions:**

Many research focuses on reduce the gap between textual and structural data representations, as I mentioned in Weakness. What are your advantages over other methods?

**Limitations:**

Yes. The limitations are discussed in Discussion section.

---

> ### Author Rebuttal · Authors · 2024-08-07
>
> Thank you for your insightful review. We acknowledge that there are many research focuses on the textual and structural data representations. However, they usually fall into the two categories mentioned in the “Introduction” section. Our structure token approach would be the third category, and this paper focuses on proving the proposed approach works under the simplest setting. Essentially, the mentioned “strong baselines with advanced graph-structured-aware methods” and our approach are not competitors. They are somewhat orthogonal and can be applied together.

---

> > ### Comment · Reviewer_eg5E · 2024-08-12
> >
> > Thanks for your response, I will keep my score.

---

### Official Review · Reviewer_3Tkr · 2024-07-10

**Soundness:** 2
**Presentation:** 3
**Contribution:** 2
**Rating:** 4
**Confidence:** 3

**Summary:**

This paper introduces TextGraphBART. Its a new method of encoding graph/text-input by using a structure token. This new token should preserve graph structure as opposed graph linearization or cycle training. This should also allow for the generation of graphs, with accompying text tokens, without making architectural changes to the transformer. It is then verified on both Graph2Text and Text2Graph tasks for commonly used benchmarks such as WebNLG and EventNarrative. It is followed by results that show comparable results to previous works, and a short discussion and conclusion.

**Strengths:**

1. Paper is generally nice to read
2. The work considers a lot of relevant related work, which they both refer to or base decisions on
3. The build-up of the structure token is well explained

**Weaknesses:**

1. The work proposes a seamless integration of both text and graph inputs, but only verifies on strictly one-directional modal evaluation: Either text-to-graph or graph-to-text. However, as the authors claim that this is the "first method that can autoregressively generate sub-graphs with multi-token labels without modifying transformers.", a single text-to-graph benchmark feels like a very shallow evaluation.
2.  Models are evaluated on datasets that already work well. This makes it hard to distinguish the added value of such a structure token. Scenarios are sketched where e.g. graph linearization is treated simply as a sequence of tokens, but then it this work should methodologically be verified on datasets where preserving graph structure should be required. This is now not the case.
3.  This is reflected in the results: they are comparable, so the structure token might aswell not be there. The proclaimed difference that this is done with less parameters I would argue is a very weak claim as the difference is not that big.
4.  Finally, I am not convinced that the proposed "structure token" is a truely lossless way of preserving graph structure through-out a transformer. The structure token and its generation is explained in Figure 2, and is claimed to be lossless. However, this is only true in the operations before the embedding leveraging the orthogonal property. After the embedding into a structure embedding, information will (seemingly) be lost about structure in this embedding. From this structure embedding, it cannot be determined what the original structure of the graph was?
5.  In the discussion the authors mention scaling, but there is no clear evidence for this as the authors only tested one model size. This is essentially an unsubstantiated claim.
 6. Why is the domain token needed? This is not further addressed or experimented with.

**Questions:**

1. Why is it that this structure token obsoletes the need for transformer modifications?
2. Why is it interesting that this method obtains comparable results?
3. Line 56: "It should also consider the cases that cannot be handled by multi-stage and graph linearization approach", so what are these cases and how are they not handled by these methods but they are by TextGraphBart?

**Limitations:**

The authors partially address this limitations by motivating the paper in the introduction. However, I dont feel the discussion (or the rest of the work) addresses the actual limitations of this method. What are pros or cons compared to transformer modifications?

---

> ### Author Rebuttal · Authors · 2024-08-07
>
> We appreciate your thoughtful review. We address your main points below:
>
> About Questions:
> 1. The necessity of transformer modifications is to incorporate the structural information of the graph into the models. TokenGT [1] shows how a good token design can obsolete the need of transformer modifications for graph data. Our structure token is built on top of TokenGT’s token design, which provides the desired structural information.
> 2. We think it is interesting even though this method only obtains merely comparable results because it shows many potential future directions in the field. The current trend of language models is extremely focused on generating sequences. Even generating structural data such as JSON data is done with sequences. Our model, while not reaching SOTA performance, still outperforms some models that are many times larger than ours. This shows the potential of rethinking how data is encoded and decoded.
> 3. Please refer to Line 35-36 and Line 48-49 for the cases not handled by previous approaches.
>
> About Weaknesses:
> 1. We test our model on both text-to-graph and graph-to-text generation. The one-directional experiments test whether the proposed method works correctly and captures the desired information. If the reviewer can suggest some tasks or datasets that require bi-directional generation, it would be some nice future explorations.
> 2. We appreciate your suggestion for using that kind of dataset. We chose to evaluate our model on these datasets because they are the common datasets for evaluating text-to-graph and graph-to-text generation models.
> 3. Please compare our model with T5-Base in Table 2 and Grapher (Query) in Table 3. If the structure token is not there, there will not be a 20% performance difference for those baselines.
> 4. The “lossless” means each text graph encoded as structure tokens can be decoded back to the original text graph, compared to the graph linearization approach. It does not refer to the neural network would be lossless.
> 5. The scaling is not a claim. We purposefully put scaling in the “Discussion” section as a limitation of the current work and explicitly state the necessity of further explorations.
> 6. Please refer to Line 128-133. This allows the model to be trained with the proposed pre-training paradigm in section 3.4
>
> [1] J. Kim, D. Nguyen, S. Min, S. Cho, M. Lee, H. Lee, and S. Hong. Pure transformers are
> 366 powerful graph learners. Advances in Neural Information Processing Systems, 35:14582–14595,
> 367 2022.

---

> ### Comment · Reviewer_3Tkr · 2024-08-12
>
> Thank you for your rebuttal. Having read it, I've decided to keep my scores.

---

### Official Review · Reviewer_XEWk · 2024-07-12

**Soundness:** 2
**Presentation:** 2
**Contribution:** 2
**Rating:** 3
**Confidence:** 4

**Summary:**

The paper proposes a new unified graph-text generation framework, TextGraphBART, for the large language model. The paper tries to address both the generation and representation of text and graphs. The paper proposes a new structure token to encode text graphs into a set of tokens. The structured token can encode the text graphs and be decoded into text graphs. Specifically, it consists of seven different embeddings, including position, domain, and text information. The paper pretrained the proposed framework over four different tasks, including text2text, graph2text, text2graph, and graph2graph. The model is pretrained on TEKGEN and GenWIKI and tests on the event narrative and WebNLG. The performance is evaluated on BLEU, METEOR< and BERTScore. The paper compares the model with T5, BART, GAP, CYcleGT, ZBT5, and Grapher. The paper also includes an ablation study and analyzes the future directions.

**Strengths:**

1. The motivation of the paper is clear and the topic of the paper is interesting. The paper investigates an important unified graph-text generation framework for the language model. The idea of structure tokens is interesting and can contribute to future research. Additionally, the model seems to achieve good results while using fewer parameters.
2. The paper tests on both graph-to-text and text-to-graph datasets. The proposed method achieves better results compared to other baselines. The paper also conducts an ablation study to investigate the contribution of each component. It includes a discussion session to explore the paper's current limitations.
3. The paper provides code and implementation details.

**Weaknesses:**

1. The proposed framework is incremental. Multiple previous papers have used the idea of using different position embeddings to represent the structure information [1,2,3]. The idea of joint text-to-graph and graph-to-text pretraining/generation is also not new [4].
2. The experiment is not comprehensive. The paper used eventnarrative for the graph-to-text generation. However, compared to WebNLG  (2020), the EventNarrative has fewer baselines and most of its baselines are outdated. The paper must also add WebNLG(2020) as an additional graph-to-text generation tool to show its contributions. The pretraining datasets and the testing datasets are all Wiki-style datasets. The performance gain may come from pretraining (data leakage) instead of the actual model architecture. The paper needs to include an additional ablation study to show the gain of the eventnarrative/webnlg comes from the model architecture, or also pretraining the proposed baselines on TEKGEN/GenWiki. Otherwise, the scores are not comparable. Additionally, the paper needs to include some of the latest frameworks like [4] since all of the baselines used in the paper are old. The paper also needs to include some human evaluation or qualitative analysis to help readers understand the generation results better. Furthermore, the comparison in Table 4 is not fair, since domain-specific pretraining is more useful than general pretraining [5].
3. The paper puts important information in the Appendix while not reaching the page limit (9 pages).

[1] Herzig, J., Nowak, P. K., Müller, T., Piccinno, F., & Eisenschlos, J. M. (2020). TaPas: Weakly supervised table parsing via pre-training. ACL 2020.

[2] Wang, Q., Yavuz, S., Lin, V., Ji, H., & Rajani, N. (2021). Stage-wise fine-tuning for graph-to-text generation. arXiv preprint ACL 2021 SRW.

[3] Chen, W., Su, Y., Yan, X., & Wang, W. Y. (2020). KGPT: Knowledge-grounded pre-training for data-to-text generation. EMNLP 2020.

[4] Wang, Z., Collins, M., Vedula, N., Filice, S., Malmasi, S., & Rokhlenko, O. (2023). Faithful low-resource data-to-text generation through cycle training. ACL 2023

[5] Gururangan, S., Marasović, A., Swayamdipta, S., Lo, K., Beltagy, I., Downey, D., & Smith, N. A. (2020). Don't stop pretraining: Adapt language models to domains and tasks. ACL 2020.

**Questions:**

See weakness

**Limitations:**

The paper includes a limitation section in the Appendix and a Discussion section in the main paper.

---

> ### Author Rebuttal · Authors · 2024-08-07
>
> Thank you for your thorough review. We address your main points below:
> 1. While the proposed framework seems to be incremental, we handle many situations that cannot be done with previous approaches. The idea of using position embeddings for structure information is well-known in graph neural networks (GNN). However, as mentioned in the “Related Work” section, these approaches are only for encoding, whereas our framework focuses on both encoding and decoding. Essentially, most of the encoding methods are not lossless encoding, which means they cannot be used for decoding. Meanwhile, our structure token is orthogonal to the joint pre-training framework, so we can apply those methods such as cycle training to our model.
> 2. About the concern of data leakage due to the wiki-style datasets, it is less of a problem because Wikipedia is also used in other pre-training models like T5/Bart. If there is data leakage, there will not be much performance difference between our model and some baselines. Furthermore, the comparison in Table 4 cannot be fair because Grapher-small (Text) is based on T5-small that pre-trained on the C4 dataset that is multiple scales larger than our TEKGEN/Genwiki pre-training dataset.
> 3. We appreciate the mention of the Appendix and the empty space (<1 page). We purposefully put the formal definitions in the Appendix for a better reading experience. We will move some of the information in the Appendix for a better use of the space in the revision.

---

> > ### Comment · Reviewer_XEWk · 2024-08-11
> >
> > Thank you for your explanation! I decide to keep my score after reading your rebuttal.

---

### Official Review · Reviewer_Njzi · 2024-07-13

**Soundness:** 1
**Presentation:** 1
**Contribution:** 2
**Rating:** 3
**Confidence:** 3

**Summary:**

- The paper highlights the limitations of two existing methods for generating text graphs: 1. The multi-stage approach does not consider multi-hop relations and cannot handle the case where two concepts have more than one relation. 2. The graph linearization approach introduces extra complexity to the Language Model (LM) and the predictions are altered if the generated triples are shuffled.
- Building on this, the paper proposes the Structure Token method. Specifically, it identifies a graph element (node or edge) using seven parts. Then, it transforms Structure Tokens into embeddings using OneHot and orthonormal-like vectors, which are then input into a Transformer Encoder-Decoder model.

**Strengths:**

- The method proposed in the paper can avoid extra computation, such as the duplication of concepts.
- The experimental results presented in the paper show that the model performs comparably to models with a larger number of parameters while using fewer parameters, suggesting that the model might perform even better with an increased number of parameters.

**Weaknesses:**

- The paper's layout is not aesthetically pleasing, particularly the formatting of Tables 1-3.
- The experiments lack error bars, which diminishes the credibility of the results.
- The experimental results are unsatisfactory and fail to demonstrate the superiority of the model, and the experiments are incomplete.
  - The model's results did not achieve state-of-the-art (SOTA) performance in the G2T/T2G tasks.
  - Lines 257-259 state, "In conclusion, our structure token approach can achieve comparable performance on text-to-graph generation under similar model size without using special training methods or loss functions." The results in Table 3 are nearly identical to those of Grapher-small (Text) T2G. Combined with the absence of error bars, it is challenging to determine whether the similarities are due to errors or model efficacy, making it difficult to convince.
  - Existing results only may demonstrate that the model performs comparably to models with larger parameters under fewer parameters. This does not prove that increasing the number of parameters will enhance performance. A more direct comparison of experimental results is necessary to substantiate this claim. The explanations in Section 5, "Scaling Up," are overly subjective and unconvincing, making the paper seem like a work in progress.

**Questions:**

- Please respond to the issues pointed out in the "Weakness" section.

**Limitations:**

Author addressed the limitations.

---

> ### Author Rebuttal · Authors · 2024-08-07
>
> We appreciate your detailed review. We address your main points below:
> 1. We appreciate the mention of the aesthetic of the layout, we will adjust accordingly in the revision.
> 2. The error bars are not included in the experiments because our model used in each experiment is finetuned from the same pre-trained weights produced by our pre-training method. There are no randomly initialized parameters during the finetuning. While we do run the same finetuning script 5 times with different random seeds, the scores are identical across 5 runs showing that the randomness in the dataset did not affect the results. The pre-trained weights will also be released with the source code.
> 3. We acknowledge that the result does not achieve SOTA performance in G2T/T2G tasks. However, our model achieves the “nearly identical” results using less pre-training data and parameters than Grapher-small (Text). Besides, our model greatly outperforms CycleGT and Grapher (Query). Combined with the good properties of our method, it demonstrates some interesting features beyond  SOTA performance. While we “anticipate” the model could scale up, we purposefully put the explanation in the “Discussion” section as a limitation of the current work and explicitly state the necessity of further explorations.

---

> > ### Comment · Reviewer_Njzi · 2024-08-11
> >
> > Thank you for explanation. I decide to keep my score.

---

### Decision · Program_Chairs · 2024-09-25

**Decision:**

Reject

**Comment:**

The paper proposes ``structure token'' to unify the representation and generation of graph and text. The reviewers are consistently negative towards the work, which could be improved by addressing the following aspects: (1) further enhancing performance to reach state-of-the-art (SOTA) levels, (2) updating the generation tool, (3) including more ablation studies, and (4) scaling up and reporting performance.